# Extracellular Vesicles: Messengers of p53 in Tumor–Stroma Communication and Cancer Metastasis

**DOI:** 10.3390/ijms21249648

**Published:** 2020-12-17

**Authors:** Evangelos Pavlakis, Michelle Neumann, Thorsten Stiewe

**Affiliations:** 1Institute of Molecular Oncology, Philipps University, 35034 Marburg, Germany; evangelos.pavlakis@imt.uni-marburg.de (E.P.); michelle.neumann@uni-marburg.de (M.N.); 2Universities of Giessen and Marburg Lung Center (UGMLC), German Center of Lung Research (DZL), Philipps University, 35034 Marburg, Germany

**Keywords:** extracellular vesicles, exosomes, p53, mutant p53, tumor microenvironment, metastatic niche priming, pre-metastatic niche

## Abstract

Tumor progression to a metastatic and ultimately lethal stage relies on a tumor-supporting microenvironment that is generated by reciprocal communication between tumor and stromal host cells. The tumor–stroma crosstalk is instructed by the genetic alterations of the tumor cells—the most frequent being mutations in the gene *Tumor protein p53* (*TP53*) that are clinically correlated with metastasis, drug resistance and poor patient survival. The crucial mediators of tumor–stroma communication are tumor-derived extracellular vesicles (EVs), in particular exosomes, which operate both locally within the primary tumor and in distant organs, at pre-metastatic niches as the future sites of metastasis. Here, we review how wild-type and mutant p53 proteins control the secretion, size, and especially the RNA and protein cargo of tumor-derived EVs. We highlight how EVs extend the cell-autonomous tumor suppressive activity of wild-type p53 into the tumor microenvironment (TME), and how mutant p53 proteins switch EVs into oncogenic messengers that reprogram tumor–host communication within the entire organism so as to promote metastatic tumor cell dissemination.

## 1. The *TP53* Tumor Suppressor in Cancer

The development of cancer is a multistep process involving a series of events that enable clonal proliferation, uncontrolled growth, and finally invasion and metastasis. Cellular stress caused by a horde of external or internal influences, such as ultraviolet radiation, hypoxia, carcinogens, oxidative stress, and oncogene activation, may lead to DNA damage and, in consequence, to the malignant transformation of the cell. In order to maintain or restore genomic integrity, the cell employs a number of damage control tools. Among them, the transcription factor p53, encoded by the *TP53* gene, functions as an exceptionally sensitive sensor of cellular stresses. It is placed at the center of a multifaceted network comprising a diverse array of p53-activating inputs and numerous outgoing effector pathways, that range from protective antioxidant and DNA repair activities to the induction of transient cell cycle arrest, permanent cell cycle withdrawal by senescence and the induction of cell death [1,2]. As such, p53 ensures that cellular homeostasis is preserved, genetic errors are prevented and oncogenic insults, responsible for inappropriate clonal outgrowth, are defeated.

Even though p53 is now considered one of our most potent tumor suppressor genes, it was originally identified in 1979 as a protein associated with the SV40 tumor-virus large T-oncoprotein [3,4], and was shown to induce malignant transformation in cooperation with oncogenic Ras [5,6,7]. It was only a decade later that the wild-type protein was revealed to suppresses malignant transformation, and that mutations, present in the first p53 cDNA clones, were responsible for the observed tumor-promoting activity [8,9,10]. Nowadays, genome-wide tumor sequencing studies have firmly established that p53’s tumor-suppressive function is abrogated in approximately 50% of all cancer cells, mostly because of missense mutations affecting up to 80% of the residues within its DNA-binding core domain [11,12]. The generation and accumulation of highly stable mutant p53 proteins (mutp53) as a result of missense mutations leads to the deregulation of a wide array of physiological signaling processes and the stimulation of tumorigenesis, not only due to the loss of p53’s normal wild-type function (loss of function; LOF), but also through a broad range of activities exerted by the mutant protein [13,14]. p53 missense mutants bind to and inactivate wild-type proteins expressed from the non-mutated allele in a dominant-negative fashion (dominant-negative effect, DNE), thus further ensuring that p53 will be disabled [15]. In addition, missense mutations are neomorphic and endow mutp53 with novel gain-of-function (GOF) properties that control tumor cell-autonomous processes, such as cancer metabolism, stemness, response to proapoptotic signals and adaptation to oxidative stress [13,14]. Consequently, gain-of-function promotes metastasis and chemoresistance, leading to decreased survival in mice and humans [14,16]. It should however be noted that although we refer to p53 mutant proteins as “mutant p53” (mutp53), more than 2000 different missense mutants have been described in cancer patients [11]. Importantly, there is substantial evidence from mice and humans that individual mutants differ not only in their structure, but also functionally, with respect to the extent of loss-of-function and dominant-negative activity, and probably even more regarding the amount and type of gain-of-function activities [16]. The p53 mutome therefore presents substantial functional diversity, forming a “rainbow of mutants” [16,17].

Although the function of p53 to constrain malignant transformation via activation of transcriptional programs is regarded primarily as a cell-autonomous response, p53’s regulatory activities extend beyond the cell membrane [18]. For example, the p53-mediated regulation of numerous genes encoding for secreted proteins enforces cellular senescence, a tumor-suppressive cellular program that involves stable proliferative arrest and microenvironmental alterations in a non-cell autonomous manner [19]. In this process, p53 closely cooperates with NF-kB, a key pro-inflammatory transcription factor, which drives the senescence-associated secretory phenotype (SASP), a term that collectively describes the upregulation of enzymes that degrade the extracellular matrix (ECM) and the secretion of a myriad of inflammatory cytokines and immune modulators, such as IGFBP-7, PAI-1, IL-6, IL-8, and CXCL-1/GRO1, many of which control, for instance, proliferation, stemness and motility in the microenvironment [19,20,21,22,23,24]. This example illustrates that it is a crucial part of p53’s tactic for resolving stress and suppressing tumorigenesis to orchestrate a coordinated tissue response, via the secretion of factors that mediate communication within the microenvironment.

Such non-cell autonomous functions of wild-type p53 are often lost, resulting in a more tumor-permissive microenvironment, when *TP53* is deleted or disrupted by nonsense or frameshift mutations, when wild-type p53 is blocked by p53 missense mutants via their dominant-negative effect, and when p53 is degraded by the expression of inhibitors, such as Mdm2 or viral oncoproteins. The neomorphic gain-of-function properties of mutp53 add additional complexity, as central secreted factors in the immediate or distant tumor microenvironment appear to be uniquely influenced by them. Prominent examples include the following: mutp53’s close cooperation with hypoxia-inducible factor 1 (HIF1) in regulating the fundamental components of the basal lamina in hypoxic tumors; the modulation of NF-kB-driven inflammatory gene expression; the hijacking of the E2F1 transcriptional program to promote the expression and secretion of pro-angiogenic cytokines; and the regulation of the proper folding of secreted N-glycosylated proteins in the ER through ENTPD5 [18].

## 2. Extracellular Vesicles—Biogenesis, Cargo and Uptake

In multicellular organisms, cells can exchange information by sending out signals composed of single molecules or, as increasingly exemplified in the literature, via complex packets stuffed with a selection of proteins, lipids, and nucleic acids, called extracellular vesicles (EVs) [25]. EVs, first detected over 50 years ago, were initially reported as cellular waste, cell homeostasis by-products or a route for the selective elimination of macromolecules form cells [26,27]. It is only in recent years that defined populations of these vesicles have been recognized as functional modes of intercellular communication, and their substantial involvement in physiological and disease-related cellular activities has begun to unravel [28]. Although EVs are highly heterogeneous in terms of their biogenesis, release pathways, physical properties and function, the generic term EVs is currently used to denote all secreted membrane-bound vesicles [29,30]. Based on today’s knowledge, two main subtypes are commonly distinguished: microvesicles and exosomes (Figure 1) [29,30]. While microvesicles (typically 100 nm to 1 µm in diameter) are formed by direct outward budding and fission of the cellular membrane, exosomes (30–150 nm in diameter) are intraluminal vesicles (ILVs) of endosomal origin [29,30]. Specifically, exosomal vesicles are generated by the inward budding of the limiting membrane of early endosomes during the maturation of the intermediates of the endosomal system and multivesicular bodies (MVBs), and are discharged via the fusion of MVBs with the cell surface [31,32]. Though the precise mechanism of exosome formation and release is still in many aspects unresolved [29], the endosomal sorting complexes required for the transport (ESCRT-I, ESCRT-II and ESCRT-III) pathway play a fundamental role, as shown by the detection of the ESCRT subunits and accessory proteins ALG-2 interacting protein X (ALIX), tumor susceptibility gene 101 (TSG101), and heat shock protein 90 β (HSP90β) in exosomes of various eukaryotic cell types [33,34,35]. However, exosomes can also be produced in an ESCRT-independent/sphingomyelinase-dependent fashion, as revealed by the presence of intraluminal vesicles, loaded with the exosomal transmembrane protein tetraspanin CD63, after the depletion of ESCRT complexes [36].

EVs are active components of the cell–cell or cell–extracellular matrix communication, and as such, they carry diverse cargo, composed of miscellaneous proteins, lipids and nucleic acids. Furthermore, while cargoes are initiators and regulators of EV formation, their nature and abundance are determined by the type of the donor cell, its state and various stimuli [28,29]. In this sense, EVs are often viewed as a miniature version of the cell of origin, while their large assortment of bioactive contents exemplifies their complexity and functional variety. For instance, in exosomes, more than 4400 proteins have been detected [37]. Among them are the following: proteins involved in the invasion and fusion processes, such as tetraspanins (CD9, CD63 and CD81); proteins related to stress responses, such as those belonging to the heat shock protein machinery (HSP70 and HSP90), as well as various multivesicular body formation proteins (e.g., ALIX and TSG101) or proteins involved in exosome biogenesis (Flotillin); cell adhesion-related glycoproteins (e.g., integrins); signaling proteins (e.g., GTPases, kinases); growth factors and cytokines (e.g., TNF-α); metabolic enzymes (e.g., pyruvate kinase, ATPases, fatty acid synthase) and many more [28,37]. Regardless of their protein content, exosomes also contain varied RNA species, the most abundant of which are microRNAs (miRNAs), which are potent regulators of gene expression. Studies showed that the miRNAs packed into exosomes frequently undergo unidirectional transfer between cells, resulting in the generation of a local trafficking network, which ultimately leads to phenotypic modifications of the recipient cells [38]. Finally, in terms of their lipid content, exosomes are enriched in fatty acids, prostaglandins and leukotrienes, while they also contain selected functional lipolytic enzymes, sufficient for the autonomous production of bioactive lipids [37].

Once released from the donor cells, EVs can reach recipient cells at local or distant sites and deliver their contents to stimulate functional responses and phenotypic modifications. Although the precise mechanisms of vesicle transfer and internalization are still not fully understood, it is believed that the target cell specificity is driven by specific interactions between proteins enriched on the EV surface and receptors on the recipient cell’s membrane. For example, integrins found on EVs can interact with intercellular adhesion molecules (ICAMs) at the surfaces of recipient cells. Even the interaction between integrins and certain extracellular matrix proteins, such as laminins and fibronectin, was shown to play an important role in the EV binding to recipient cells [39]. Other mechanisms of EV–target cell interactions include their uptake either by the phagocytosis or fusion of EV membranes with the target cell’s plasma membrane [40,41].

Of note, the incorporation of cargo into EVs is considered a non-random event, and the regulated mechanisms of selective cargo loading have been described. Even though the detailed mechanisms are still poorly understood, protein sorting into EVs is strongly affected by the mode of vesicle biogenesis, and is dynamically controlled by post-translational modifications such as ubiquitylation, sumoylation or farnesylation [29,42]. Moreover, different machineries have been proposed to perform specific nucleic acid sorting activities, including the ESCRT‑II subcomplex that could act as an RNA-binding complex [43], the tetraspanin-enriched microdomains that could sequester RNA-binding proteins in the membrane subdomains [44], or the miRNA-induced silencing complex (miRISC) and protein argonaute 2 (AGO2), which mediates the RNA silencing processes [29,45,46]. Different pathways of biogenesis and cargo sorting thereby generate a diverse set of EV subtypes with a fine-tuned cargo load. On the one hand, regulated cargo sorting might be a mode for the selective disposal of unnecessary cell components. On the other, it likely represents a dedicated mechanism for communicating dynamic signals to the microenvironment in order to ensure homeostasis at the tissue or organismal level.

## 3. Extracellular Vesicles—Function in Cancer

Apart from roles in homeostasis, EVs have been identified as pathogenicity factors in a wide range of diseases, including cardiovascular disease, neurodegeneration and cancer [47]. In particular, the study of exosomes in cancer has progressed at a rapid pace compared with research into their roles in other diseases, and exosomes have been associated with most of the hallmark features of cancer [47]. Numerous studies have revealed that the EVs secreted by cancer cells transmit signals not only to neighboring cancer cells, but also to various host cells. Tumor-released EVs impact directly on tumor stroma cells in the primary tumor’s microenvironment and, following their dissemination via blood circulation, initiate the establishment of metastatic niches in distant organs [48]. Processes such as the activation of proliferative and angiogenic pathways, the silencing of tumor-suppressive signals and the reprogramming of the immune surveillance machinery, have all been demonstrated to be influenced by homotypic or heterotypic intercellular communication through EVs [49,50,51].

### 3.1. EVs Shape a Tumor-Supporting Microenvironment

A prime component of the tumor microenvironment of most solid tumors is cancer-associated fibroblasts (CAFs), a heterogeneous and plastic cell population comprising, in the simplest view, cells negative for epithelial, endothelial and leukocyte markers with an elongated morphology and lacking the mutations found within cancer cells [52]. CAFs play a central role in tumor progression and metastasis by the production of growth factors and the synthesis and remodeling of the extracellular matrix [53]. In addition, CAFs modulate the immune system and influence angiogenesis, tumor mechanics, drug access and therapy responses [52]. CAF-derived EVs have been identified as key mediators in many of these processes [54]. For example, EVs released from prostate CAFs were shown to inhibit mitochondrial oxidative phosphorylation, thereby increasing glycolysis and glutamine-dependent reductive carboxylation in cancer cells, and to shuttle metabolites into cancer cells promoting tumor growth under nutrient stressed conditions [55]. EVs from breast CAFs were shown to contain and transfer mtDNA to cancer stem cells, promoting estrogen receptor-independent oxidative phosphorylation, escape from metabolic dormancy and endocrine therapy resistance [56]. Pancreatic ductal adenocarcinoma-associated CAFs are intrinsically resistant to the commonly used chemotherapeutic drug gemcitabine, and, under chemotherapy, increase the release of EVs that promote the proliferation and survival of the recipient epithelial tumor cells [57]. In addition, CAFs containing a specific miRNA signature have been critically involved in colorectal cancer (CRC) development and expansion, and by releasing exosomal Wnt induce the dedifferentiation of tumor cells and promote drug resistance [58,59].

Another principal component of the tumor microenvironment is the immune cell compartment, which during tumorigenesis undergoes extensive editing by malignant cells to evade the immune response and create a well-balanced, proinflammatory environment, in which a network of distinct immune cell subtypes cooperatively drives tumor progression. EVs were shown to be instrumental in establishing and maintaining this immune-privileged environment. For example, exosomes derived from mouse breast cancer cells contribute to tumor expansion by blocking the IL-2 mediated proliferation and activation of natural killer (NK) cells [60]. Exosomes from glioblastoma cells stimulate the differentiation of monocytes into immunosuppressive macrophages [61]. The establishment of an immunosuppressive environment may also be mediated by the release of TGF-β/miR-23a-containing EVs from hypoxic tumors, which induce regulatory T cell (Treg) activity and inhibit NK cell cytotoxicity [62].

The growth of tumors to a clinically detectable size is metabolically limited by nutrient and oxygen diffusion, and requires the stimulation of neoplastic angiogenesis, a process which is also stimulated by cancer cell-secreted EVs. For instance, colorectal cancer-derived exosomes with defined cell cycle-related mRNA signatures, EVs released from glioblastoma tumors under hypoxic conditions, or breast cancer-secreted EVs containing miR-210, can all promote vascular endothelial cell proliferation and induce neoangiogenesis [63,64,65]. Furthermore, the exosome-mediated transfer of miR-105 from tumors to the endothelium leads to the destruction of endothelial cell barriers, thus facilitating cancer cell access to the blood stream for dissemination and metastasis [66].

### 3.2. EVs Stimulate Metastasis and Prime Metastatic Niches

In addition, EVs operate during the metastatic spread in tumor cell migration and invasion of the surrounding tissue. These activities often involve a bidirectional crosstalk between tumor and stroma cells. For example, colorectal cancer cells release miR-1246-enriched exosomes that trigger macrophage reprograming to a tumor-supporting, M2-polarized subtype [67]. In turn, such tumor-associated macrophages release EVs containing miR-21-5p and miR-155-5p that increase the migration and invasion of colorectal cancer cells [68]. Likewise, exosomal miR-106b-3p is abundantly detected in the serum of metastatic colorectal cancer patients, induces tumor cell migration, invasion and epithelial-to-mesenchymal transition (EMT), and correlates with poor survival [69,70]. Other pro-invasive factors transported via tumor-released EVs include TGF-β, caveolin-1, HIF1α and β-catenin, all of which were shown to promote EMT, extracellular matrix remodeling and metastatic niche formation [71]. Examples come from multiple cancer types: colorectal cancer, where EVs transfer mutated β-catenin to modulate Wnt signaling and enhance tumor growth [72]; gastric cancer, where EV-mediated TGF-β transfer and subsequent activation of the TGF-β/Smad pathway assists in the formation of metastatic niches [73]; and nasopharyngeal carcinoma, where EVs with enhanced levels of HIF1α increase the migration and invasion of tumor cells [74].

One of the most interesting conceptual advances in the field of cancer metastasis is the realization that organs of future metastasis are not passive receivers of circulating tumor cells, but are instead selectively and actively modified by the primary tumor before metastatic spread has even occurred [75]. In this process, EVs are released by the primary tumor into the circulation and act as long-distance messengers to instigate hospitable and immune-privileged pre-metastatic niches in distant organs that facilitate future colonization by circulating tumor cells [76]. In mice, the administration of exosomes derived from highly metastatic melanoma cells, prior to subcutaneous transplantation with melanoma cells, promoted metastasis to lung and bone [77]. The exosomes transferred the receptor tyrosine kinase MET to bone marrow progenitor cells, educated them toward a pro-vasculogenic and pro-metastatic phenotype and recruited them to pre-metastatic sites through the upregulation of proinflammatory molecules [77]. Vice versa, exosomes from poorly metastatic melanoma cells reduce the metastatic burden of highly metastatic primary tumors, suggesting that nonmetastatic exosomes may prevent metastatic disease [77]. In pancreatic cancer, exosomes direct the formation of pre-metastatic niches in the liver [78]. Specifically, pancreatic ductal adenocarcinoma-derived exosomes containing macrophage migration inhibition factor (MIF) are taken up by Kupffer cells, which in turn induce the formation of a fibrotic environment via the production of fibronectin from hepatic stellate cells and the stimulation of TGF-β secretion [78]. The altered microenvironment allows the enhanced recruitment of bone marrow-derived macrophages, and the establishment of liver pre-metastatic niches and finally metastases [78]. Interestingly, exosomes not only affect the extent of metastasis, but can also direct metastasis to specific organs [39]. Mechanistically distinct integrin expression patterns on tumor-released exosomes define the cell types with which the vesicles will fuse at distant sites, thereby determining organotropic spreading to lung, liver or brain [39].

## 4. EVs as Extracellular Messengers of p53

Cancer phenotypes are primarily instructed by the genetic alterations that cause tumor development. It is therefore not surprising that p53 as the most frequently mutated cancer gene affects how tumor cells communicate through EVs (Figure 2). In the following sections we therefore review, first, how wild-type p53 controls and utilizes EV-mediated cell communication, primarily to extend its homeostatic and tumor-suppressive functions from the inside of the cell into the cellular microenvironment. These EV-mediated activities of wild-type p53 are typically lost when p53’s function becomes compromised in tumor cells by *TP53* gene mutations or inhibitory signals. However, as many tumors contain *TP53* missense mutations that produce highly stable mutant proteins with neomorphic gain-of-function properties, we review in a second section how p53 mutants employ EVs in a novel, and often mutation-specific, manner, as oncogenic messengers that reprogram tumor–host communication to promote cancer progression.

### 4.1. EVs Communicate Tumor Suppressive Signals of Wild-Type p53

Several decades of research have established that p53’s unique ability to provide a barrier to malignant transformation lies within the core of its most central function, which is to sense stress inputs and to coordinate complex effector pathways that restore cellular homeostasis and maintain genomic stability. Besides operating primarily within cells, the activation of the p53 protein orchestrates molecular programs leading to an increase in the export of growth-suppressive factors that spread p53’s tumor suppressive action into the extracellular space (Figure 3) [18].

#### 4.1.1. p53 Stimulates EV Secretion

One of the earliest studies demonstrating this in vitro and in vivo noted a p53-dependent secretion of factors induced by ionizing radiation (IR), comprising the anti-angiogenic protein thrombospondin and the anti-cancer serine protease inhibitor plasminogen activator inhibitor-2 (PAI-2) [85]. Importantly, the secretion and accumulation of these factors caused a “bystander effect” of growth inhibition in a number of cells, suggesting that the induced export of growth suppressive stimuli from damaged to neighboring cells potentiates a well-coordinated non-cell-autonomous p53-regulated stress response [85]. Bystander phenomena as a means of cellular communication are mediated via the transfer of information through gap junctions, extracellular matrix remodeling or the release of soluble factors, and have sporadically been portrayed in tumorigenesis as well as in tumor therapy [86,87,88]. However, it was not until the role of EVs in such processes was exposed that dots began to connect. For instance, the activation of p53 by IR was found to result in a dramatic alteration of the protein secretome, prominently elevating the extracellular abundance of proteins such as HSP90β, EF1α, maspin, PGK-1, PAI-I and PRDX-1 [79]. Surprisingly, many of these factors are not encoded by p53 target genes and do not contain the NH2 signal sequence of secreted proteins, and therefore do not fulfill the classical rules for secretion. Instead, these proteins were observed to exit the cell as exosomes, and it was the exosome production itself that was induced by the p53 response [79]. Similarly, more recent studies also demonstrated that exosomes derived from UV-irradiated cancer cells show altered RNA content, and trigger mitochondrial depolarization and the death of non-irradiated cells [89]. 

Mechanistically, p53 controls EV secretion at multiple levels. First, in response to stress p53 transactivates the tumor suppressor activated pathway-6 (*TSAP6*) gene, encoding a 5-6 transmembrane protein that facilitates cell cycle arrest and apoptosis [90]. However, TSAP6 was also found to interact with the histamine-releasing factor (also known as TCTP) and increase its exosomal secretion, suggesting a p53-inducible role in the selective transport of proteins to the exosome, or a more general role in regulating exosome production [91]. In line with the role of TSAP6 as a mediator of DNA damage-inducible exosome secretion downstream of p53, the overexpression of TSAP6 facilitates the production of exosomes independent of p53 status [79]. This notion is reinforced by the severe in vivo defects in DNA damage-induced p53-dependent exosome secretion in TSAP6-deficient knockout mice [92]. Intriguingly, TSAP6 knockouts show a diminished p53-dependent apoptotic response to γ-irradiation in the spleen [92], suggesting that p53-TSAP6-dependent exosome secretion contributes to the exceptionally high radiation sensitivity of hematopoietic tissues. How TSAP6 increases exosomal protein secretion has not yet been delineated. Nevertheless, *TP53* gene mutations in cancer cells are expected to compromise or ablate these effects of wild-type p53 on TSAP6-dependent exosome production, and TSAP6 expression alone is sufficient to restore DNA damage-inducible exosome secretion to p53-null cells [79]. Notably, studies from colorectal cancer patients, however, found no correlation between tissue p53 protein expression, TSAP6 (mRNA and protein) and plasma exosome levels, indicating that perhaps different mechanisms exist in different cancer entities [93].

In addition to TSAP6 induction, p53 also profoundly impinges on the intracellular vesicle trafficking system by regulating basic components of the endosomal compartment. Endosomes have a number of functions in cells, ranging from the internalization of the membrane proteins and receptors into multivesicular bodies, to the production of exosomes and autophagic vesicles. It is in the course of this process that choices are made on whether cargo will be trafficked outside the cells via EVs or into lysosomes for degradation and autophagic vesicles. p53 directly transactivates expression of the ESCRT-III subunit CHMP4C [80]. ESCRT-III consists of oligomers or polymers of small α-helical CHMP proteins, of which CHMP4 paralogues are the most abundant components. ESCRT-III is recruited by ESCRT-II and constitutes the key functional component in driving membrane constriction. This is achieved through the formation of membrane-binding spirals that mediate membrane deformation and scission, in cooperation with the AAA-ATPase VPS4 [94]. This canonical ESCRT pathway can be intersected by syntenin and the ESCRT accessory protein ALIX (also known as programmed cell death 6-interacting protein), which bridge cargoes and the ESCRT-III subunit CHMP4 [29,95]. In addition, p53 transactivates Caveolin-1 (CAV1), one of the main constituents of the Caveolae membrane vesicles responsible for the internalization of membrane receptors such as EGFR [80]. As such, the functions of the endosome compartment, exosome biogenesis, endosome production, receptor internalization and recycling increase as a result of the DNA damage-induced p53 response [80]. It is believed that these mechanisms enable p53 to better shut down cell growth and division, and communicate stress signals to other cells in the microenvironment so as to elicit a coordinated homeostatic tissue response [80].

#### 4.1.2. p53 Controls EV Size

Another study comprehensively profiled exosomal proteins released from parental, wild-type p53-expressing HCT116 cells versus p53-knockout derivative cells, or cells transfected with the p53 R273H mutant [81]. Among more than 140 deregulated proteins, the hepatocyte growth factor-regulated tyrosine kinase substrate (HGS, also known as HRS or VPS27) stands out as it was consistently increased in the exosomes secreted from wild-type compared to p53 mutant or null cells [81]. Moreover, in colorectal cancer patients, the mutational loss of wild-type p53 function correlated with reduced HGS mRNA levels [81]. HGS forms a heterotetrameric, multivalent ubiquitin-binding complex with STAM, often referred to as ‘ESCRT-0’, and mediates the entry of ubiquitylated cargoes into the intraluminal vesicles of the multivesicular body [94]. Consistent with its role in exosome biogenesis, HGS levels were previously shown to determine the number and size of the secreted vesicles, which is dependent on the amount of loaded cargo [96]. Interestingly, EVs from both p53-deficient and p53-mutant HCT116 cells were smaller in size, and this effect could be recapitulated by HGS depletion in parental, p53 wild-type HCT116 cells, suggesting that the p53-HGS axis might control exosome size [81]. Nevertheless, given the lack of in vivo data, the significance of these findings in tumor progression still remains elusive.

#### 4.1.3. p53 Promotes EV Loading with Anti-Metastatic Cargo

Although EVs have frequently been shown to draw on both ends of the equilibrium that defines tumor suppression and tumorigenesis, the precise mechanisms behind the proclivity towards one direction or the other have remained obscure [77,97,98,99]. A recent mouse model of experimental melanoma metastasis identified the chaperone Bcl-2-associated anthogene 6 (BAG6) as a molecular toggle that determines the secretion of EVs with anti- or pro-metastatic cargo [82]. BAG6 is a regulator of T- and NK-cell activity and an essential co-factor for the CBP/p300 acetyltransferase which acetylates histones and p53 [100,101,102]. Intriguingly, EVs derived from BAG6-deficient and BAG6-containing melanoma cells differ significantly in their protein and mRNA cargo, and mice conditioned with these EVs develop distinct molecular changes in their lungs [82]. On one hand, EVs from BAG6-proficient cells promote the expression of anti-tumorigenic factors, such as the metalloproteinase inhibitor 3 (TIMP3), interleukin 10 (IL10) or chemokine CXCL13, and give rise to the accumulation of anti-tumorigenic patrolling monocytes [82]. On the other hand, BAG6-KO EVs stimulate the upregulation of transcripts associated with the accumulation of neutrophils, which are key players in the formation of pre-metastatic niches [82,103]. Intriguingly, the recruitment of neutrophils or tumor-promoting macrophages at distant organ sites is considered to be a defining moment for the EV-dependent promotion of metastasis [78,98]. Mechanistically, the BAG6-dependent release of anti-tumor EVs was found to require direct BAG6 association with the CBP/p300 acetylase, and subsequent translocation into the cytoplasm to acetylate p53 [82,84]. Of note, the nuclear import of BAG6 and CBP/p300 in response to starvation triggers the acetylation of nuclear p53 and autophagy [104]. Given that autophagy is often considered mutually exclusive with the release of exosomes [105], this suggests a key role for BAG6 nucleo-cytoplasmic shuttling in controlling CBP/p300 availability in the two compartments, and switching p53 functions between autophagy and exosome production. Like p53, BAG6 is also more directly involved in EV biogenesis. Upon DNA damage, BAG6 co-immunoprecipitates with components of the ESCRT machinery, in particular HGS, ALIX and TSG101, which has been shown to be dependent on its late endosomal motif P(S/T)AP [82]. In view of all this, future investigations are expected to shed more light on the BAG6/CBP/p300-p53 axis and how it is affected by pro-metastatic p53 mutations.

#### 4.1.4. p53 Suppresses the Generation of Cancer-Associated Fibroblasts through EVs

Exosomes released from wild-type p53 cancer cells are not only instrumental in recruiting tumor-suppressive immune cells, but also limit the activation of fibroblasts into cancer-associated fibroblasts that stimulate cancer growth and metastasis through the synthesis and remodeling of the extracellular matrix and the production of pro-angiogenic growth factors [52]. The loss or mutation of p53 in colorectal cancer cells was shown to stimulate via exosomes the proliferation and tumor-supporting function of co-cultured fibroblasts by reducing their p53 expression [83]. Mechanistically, these pro-tumorigenic, fibroblast-stimulating effects of the exosomes released by wild-type p53-deficient colorectal cancer cells are explained by an upregulation of multiple exosomal miRNAs, including miR-1249-5p, miR-6737-5p and miR-6819-5p, which all target the *TP53* mRNA [83]. Of note, the intracellular expression levels of these miRNAs were not increased by p53 depletion, suggesting a wild-type p53-controlled mechanism that decreases the sorting of selected miRNAs into exosomes.

#### 4.1.5. p53 Supports Chemoimmunotherapy Responses through EVs

Cancer progression and metastasis is associated with the development of therapy resistance, which is often dictated by complex interactions between malignant cells and the immune compartment. In a recent study on the chemoimmunotherapy of B cell malignancies, wild-type p53 was found to play a crucial role in the successful phagocytic elimination of cancer cells by macrophages [84]. As shown in primary patient samples with wild-type p53 status, the combination treatment with chemotherapy and anti-CD20 antibodies results in highly antibody-dependent cellular phagocytosis, [84]. This effect is lost as a result of p53 mutations, indicating that wild-type p53 modulates the communication between cancer cells and the tumor microenvironment by driving the phagocytic capacity of tumor-fighting macrophages [84]. In this scenario, p53 operates as a functional switch that suppresses the release of EVs, which transfer immune checkpoint molecules, in particular programmed-death ligand 1 (PD-L1) [84]. Inactivation of p53 not only results in the secretion of increased numbers of EVs, but also in EV cargo alterations, and particularly, the elevated expression of vesicular PD-L1, which drives resistance to therapeutic targeting and can be overcome with an anti-PD-L1 immune checkpoint inhibitor [84].

### 4.2. EVs Communicate Tumor-Promoting Signals from Mutant p53

As *TP53* is most frequently affected by missense mutations that not only disrupt the wild-type p53 functions described above, but often simultaneously create neomorphic gain-of-function activities that operate in metastasis and therapy resistance, it is conceivable that mutant p53 proteins actively reprogram EV-based tumor cell communication. Indeed, in the last few years several reports have revealed that mutant p53 modulates the EV secretome, with remarkable effects on the composition of the microenvironment at the primary tumor and even at sites of future metastasis (Figure 4).

#### 4.2.1. Mutant p53 Effects on EV miRNA Cargo

In one of the first of these studies, mutp53 in colorectal cancer cells was shown to initiate through EV cargo alterations a positive feedback with macrophages that promotes tumor growth and metastasis [67]. In detail, colorectal cancer cells expressing mutp53 were found to secrete exosomes selectively enriched in several miRNAs, including miR-1246. Of note, the miR-1246 expression in the tumor cells was not altered by p53 status. Instead, mutp53 was found to increase miR-1246 sorting into exosomes by promoting the SUMOylation of the RNA-binding protein hnRNPa2b1, which binds to specific motifs in miRNAs and drives their sorting into vesicles [109]. Eventually, miR-1246-enriched exosomes are taken up by adjacent macrophages leading to their reprogramming into an anti-inflammatory, tumor-supportive M2-like phenotype characteristic of tumor-associated macrophages (TAMs) [67]. Consistently, the immunohistochemistry of colorectal cancer samples revealed a prominent presence of TAMs at the invasive front of the mutp53 tumors. Moreover, macrophages co-cultured with p53-mutant colorectal cancer cells demonstrate enhanced extracellular matrix degradation, possibly due to the increased motile and invasive properties caused by reprogramming. Along with changes in cell migration and invasion, the miR-1246-dependent reprogramming of macrophages induces secretion of, among other factors, IL-10, TGF-β, and matrix metalloproteinases, generating an anti-inflammatory microenvironment, the recruitment of immunosuppressive Tregs, the epithelial-to-mesenchymal transition of tumor cells and increased tumor growth and metastasis. When wild-type p53 tumor cells were subcutaneously injected into mice, together with macrophages that had been previously co-cultured with mutp53-expressing colon cancer cells, tumors grew faster, with increased metastatic burdens for the lung and liver. Importantly, these observations were recapitulated with macrophages transfected with miR-1246 mimics and blocked by the miR-1246 inhibitor, identifying miR-1246 as the responsible exosomal cargo. While this study highlights an intriguing non-cell-autonomous mechanism whereby mutp53 cancer cells promote tumor growth and metastasis through the active modulation of immune cells within the tumor microenvironment, it immediately raises the question of whether this exosome-mediated crosstalk is exclusive for macrophages, whether other (immune) cells in the tumor stroma are also affected by exosomal miR-1246, and whether other mutp53-enriched exosomal miRNAs play a similar or distinct role.

The concept of an exosome-mediated mutp53 gain-of-function in the tumor microenvironment was further extended by a recent study demonstrating the mutp53-derived activation of stromal fibroblasts, which is also dependent on exosomal miRNA transfer, and similarly enhances the pulmonary metastasis of colorectal cancer cells [106]. Here, the introduction of the p53 gain-of-function mutant R273H into the (wild-type p53) colorectal cancer cell line HCT116 induced changes in the miRNA composition of tumor cell-derived exosomes that lead to the activation of co-cultured mouse embryonic fibroblasts (MEFs). In particular, exosome-exposed MEFs upregulated α-SMA, vimentin, and TGF-β expression. Microarray analysis revealed a mutp53-induced increase in exosomal miR-21-3p and miR-769-3p expression. Both miRNAs synergistically trigger myofibroblast activation and correlate with p53 mutations in the TCGA cohort of colorectal cancer patients. Of note, Smad7, an inhibitor of the TGF-β pathway and target of miR-21-3p, was inversely correlated with p53 mutations in the colorectal cancer patient cohort, providing a potential mechanistic explanation for the activation of cancer-associated fibroblasts. Moreover, TGF-β is not only an activator of CAFs, but is also a potent inducer of the epithelial–mesenchymal transition (known as EMT). EMT is associated with the disassembly of cell–cell junctions, cell detachment, the upregulation of matrix metalloproteinases and stem cell markers, and it ultimately leads to a migratory and invasive tumor cell phenotype with high metastatic potential [110,111]. EMT is therefore considered central to the metastatic spreading of carcinoma cells [112]. In line with this, mutp53 colorectal cancer xenografts showed upregulation of mesenchymal markers and increased pulmonary metastasis. As wild-type p53 reinforces the epithelial phenotype in a cell-autonomous manner, for example through the upregulation of miR-34 and miR-200 family members [113,114,115], it is intriguing to speculate that mutant p53 establishes a non-cell-autonomous tumor–stroma crosstalk, driven by exosomal miRNA secretion, that enhances TGF-β secretion by fibroblasts, to reinforce the pro-metastatic EMT phenotype of colorectal cancer cells.

#### 4.2.2. Mutant p53 Effects on EV Protein Cargo

Apart from stimulating the exosomal transfer of miRNAs, mutp53 was also shown to confer pro-migratory and pro-invasive attributes to tumor and non-tumor cells by altering the exosomal expression levels of podocalyxin (PODXL), a sialomucin and glycocalyx component that functions as a transmembrane adhesion receptor associated with cancer aggressiveness [107]. The authors had previously shown that mutant p53 increases tumor invasiveness in a cell-autonomous manner, involving mutp53 inhibition of the p63/DICER/miRNA pathway, by upregulating the Rab-coupling protein (RCP) and the diacylglycerol kinase-α (DGKα)-dependent endosomal recycling of integrins and receptor tyrosine kinases [116,117,118]. Integrins are continuously endocytosed and recycled back to the plasma membrane, and the relative rate at which integrins are trafficked through the endosomal pathway is essential for the migratory behavior of cells [119]. Alterations in integrin trafficking can modulate migration, which is reflected by a switch between persistent/slow and random/rapid cell movement [120]. Interestingly, the authors noted that the mutp53 effect—but not the mutp53 protein itself—can be transferred by exosomes to influence RCP- and DGKα-dependent integrin trafficking and cell migration in p53-null cells. Comparative proteome analysis of mutp53 and p53-null cells identified PODXL to be suppressed in exosomes shed from mutp53 cells. The reduced sorting of PODXL into exosomes is caused by the transcriptional downregulation of PODXL, dependent on mutp53’s ability to inhibit p63, and involves Rab35 GTPase which binds to PODXL and influences its sorting into exosomes. The uptake of mutp53 cell-derived exosomes enhances integrin trafficking also in fibroblasts, stimulates their migratory behavior and, probably even more importantly, causes remodeling of the extracellular matrix into a more branched, orthogonal network that is more permissive to invasion. Intriguingly, these exosome-mediated effects also seem to operate over long distances, as mutp53-expressing but not p53-null subcutaneous xenografts as well as p53 mutant autochthonous pancreatic tumors triggered pro-invasive extracellular matrix alterations in the lung that could serve as pre-metastatic niches to facilitate metastatic colonization.

The heat shock protein HSP90α is another protein that is secreted via exosomes in a mutp53-dependent manner [108]. In many *TP53*-mutated tumor cells, mutp53 is stabilized by the HSP90 chaperone machinery, which prevents degradation by the E3 ubiquitin ligases Mdm2 and CHIP [121,122]. Given the addiction of many tumor cells to the neomorphic gain-of-function activities of mutp53 [123], HSP90 is considered a promising target for anti-cancer therapy, and HSP90 inhibitors that trigger mutp53 degradation are currently tested in clinical trials [17,124,125]. In addition, HSP90 also exists in an extracellular form (eHSP90α) that positively correlates with tumor malignancy, tumor cell motility and metastasis [126,127]. eHSP90α is secreted in response to certain stress stimuli by several tumor entities, and can interact with, among others, matrix metalloproteinase-2, and promote its stabilization and activation with the result of extracellular matrix degradation and increased tumor cell dissemination [128]. The secretion of eHSP90α was found to be exosome-mediated and regulated by p53 status; it is suppressed by wild-type p53, increased in the absence of p53, and further boosted selectively by DNA contact mutants (R273H, R280K), but not by a conformation mutant (R175H) [108]. While both types of p53 mutants stimulate migratory and invasive behavior in vitro and increase metastasis in mouse models, only the pro-metastatic effect of the DNA contact mutant is dependent on eHSP90α, and is neutralized by HSP90α antibodies [108]. Exosome-mediated eHSP90α secretion is associated with the translocation of cytoplasmic HSP90α to the plasma membrane [129]. Detailed analysis indicated that mutp53 expression or the loss of wild-type p53 enhances HSP90α trafficking and secretion via the vesicular trafficking pathway, but does not affect exosome biogenesis and release, suggesting that mutp53 primarily accelerates the process of HSP90α sorting to exosomes. Furthermore, HSP90α was found to co-immunoprecipitate with RCP, which sorts cargo proteins to Rab11A+ endosomes for vesicular trafficking [120]. The HSP90α–RCP complex is reduced by wild-type p53 and selectively stimulated by R273H (not R175H), which mirrors changes in overall RCP mRNA and protein expression. The invasion- and metastasis-promoting effects of mutp53 are abrogated by RCP depletion and rescued with recombinant HSP90α, confirming the existence of a pro-invasive pathway involving exosomal HSP90α-secretion stimulated by mutp53 via RCP [108]. Remarkably, the proinvasive functions of the DNA contact mutants (R273H and R280K) are more dependent on eHSP90α than those of the conformational mutp53 (R175H), explained by their different abilities to stimulate exosomal HSP90α secretion [108].

## 5. Conclusions

*TP53* mutations are indisputably recognized as drivers of cancer progression to a more aggressive, metastatic and therapy-resistant state. Although this is sufficiently explained by numerous cell-autonomous mechanisms, results from recent years provide compelling evidence that non-cell-autonomous activities reinforce the intracellular functions of wild-type and mutant p53 by communicating signals to the microenvironment that elicit coordinated multicellular responses. In this respect, *TP53* mutations, resulting in loss of wild-type p53 function and/or additional novel oncogenic properties, bring about widespread changes in reciprocal tumor–host communication by controlling the secretion of extracellular messengers such as EVs and vesicle-bound proteins. p53 is dynamically involved in the core of the EV biogenesis pathway, and *TP53* mutations not only abrogate this effect directly or indirectly via a dominant-negative action, but they even exert additional gain-of-function activities that further favor tumor progression.

In principle, many of these p53-imposed alterations on tumor cell EV secretion could be used clinically for diagnostic purposes, as prognostic biomarkers or, ideally, as novel targets for the treatment of the particularly therapeutically challenging class of p53-mutant cancers. However, many of these described mechanisms have so far only been delineated for a small number of hotspot mutations or for specific cancer types. Notably, among the still-limited number of studies, examples have surfaced wherein different p53 mutants utilize distinct EV-based mechanisms to acquire oncogenic activities [108]. Considering the tremendous complexity of the *TP53* mutome with its rainbow of more than 2000 functionally diverse mutants, it will certainly remain a challenging task to pinpoint p53-dependent changes in EV secretion that are clinically relevant and characteristic for a larger number of p53 mutants across a broad range of cancer entities.

## Figures and Tables

**Figure 1 ijms-21-09648-f001:**
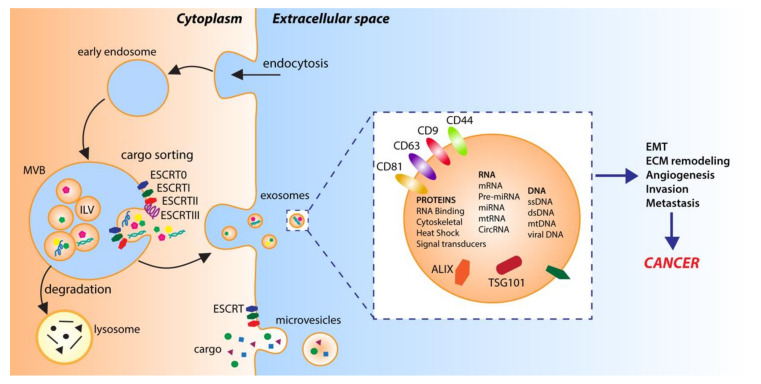
Extracellular vesicle biogenesis. Microvesicles are generated by direct outward budding from the plasma membrane. Exosomes are derived from intraluminal vesicles (ILV) generated by inward budding of the limiting membrane of early endosomes and multivesicular bodies (MVB). Endosomal sorting complexes required for transport (ESCRT0-III) and RNA-binding proteins are involved in sorting cargo into ILVs.

**Figure 2 ijms-21-09648-f002:**
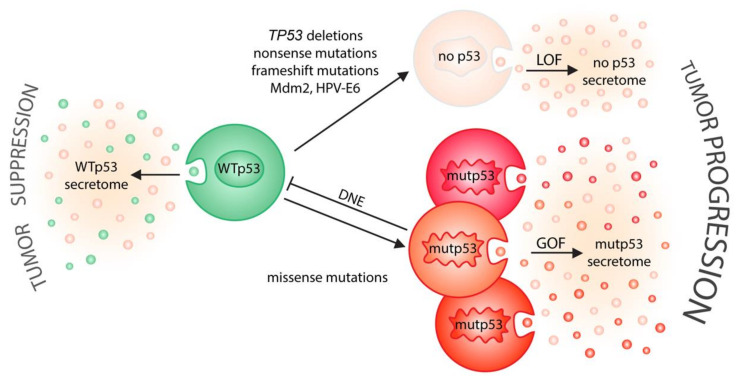
p53 status shapes tumor–host communication via extracellular vesicles. On one side (left), wild-type p53 induces the secretion of EVs that relay tumor suppressive signals to the microenvironment. On the other side (right), cancer-associated alterations in p53 status drive tumor progression by neutralizing the secretory effects of wild-type p53 (LOF, loss-of-function; DNE, dominant-negative effect) and/or by creating a mutant p53-specific EV secretome (GOF, gain-of-function) that is likely different for distinct missense mutations.

**Figure 3 ijms-21-09648-f003:**
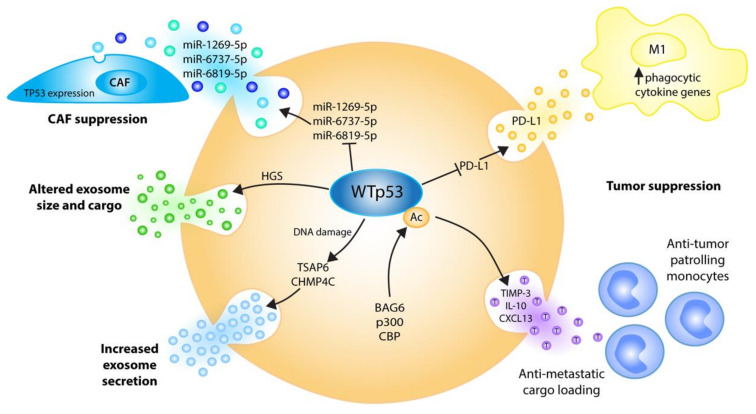
Wild-type p53 regulates the EVs’ biogenesis, cargo and cancer functions at multiple levels to communicate tumor-suppressive signals to the microenvironment. Stress-induced p53 transactivates TSAP6 and the ESCRT-III subunit CHMP4C, resulting in increased exosome secretion [79,80]. p53 alters exosome size and cargo via ESCRT-0 subunit HGS [81]. DNA damage-triggered p53 acetylation by BAG6/CBP/p300 induces anti-metastatic EV cargo, causing the recruitment of tumor-suppressive patrolling monocytes to sites of future metastasis [82]. p53 suppresses cancer-associated fibroblasts’ (CAF) proliferation by repressing exosomal *TP53*-targeting miRNAs [83]. p53 supports tumor cell phagocytosis under chemoimmunotherapy by repressing the release of PD-L1-positive EVs [84].

**Figure 4 ijms-21-09648-f004:**
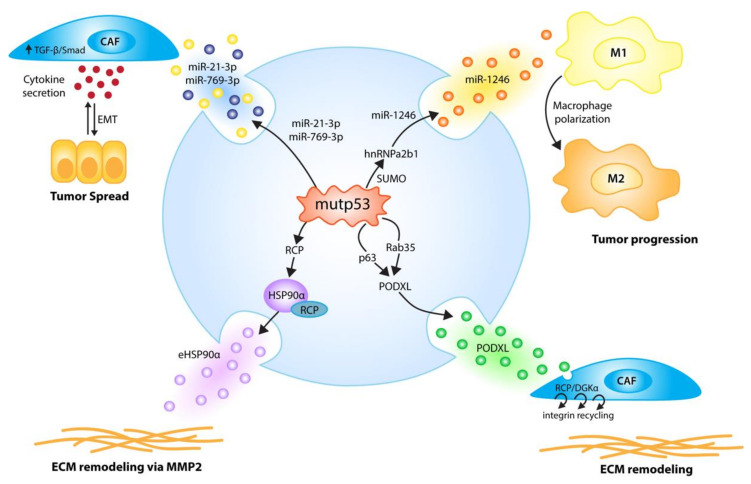
p53 mutants (mutp53) employ EVs to communicate signals to the host microenvironment that promote tumor growth and metastasis. p53 mutants induce the secretion of EVs enriched in selected miRNAs; miR-1246-enriched EVs, due to hnRNPa2b1 SUMOylation, reprogram macrophages to a tumor-supporting M2-like phenotype [67], while the mutp53-induced secretion of miR-21-3p/miR-769-3p EVs activates cancer-associated fibroblasts (CAFs) to secrete cytokines and enhance tumor cell motility via an epithelial-to-mesenchymal transition (EMT) [106]. Via p63 and Rab35, p53 mutants fine-tune exosomal PODXL levels, thereby increasing RCP/DGKα-dependent, pro-invasive integrin recycling in recipient tumor cells and fibroblasts, which leads to enhanced invasion and creates pro-invasive extracellular matrix (ECM) alterations in distant organs resembling pre-metastatic niches [107]. p53 DNA contact mutants, via RCP, increase exosomal HSP90α secretion, leading to pro-metastatic extracellular matrix remodeling via MMP2 [108].

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
