# Peer review of "Extracellular Vesicles: Messengers of p53 in Tumor–Stroma Communication and Cancer Metastasis"

_ijms, 2020, doi:10.3390/ijms21249648_

Round 1
Reviewer 1 Report
The review paper does an excellent job in demonstrating how p53 proteins control secretion, size and especially RNA, and protein cargo of tumor-derived EVs. The Abstract is well written and balanced and quickly helps the reader to get a gist of the core topic in discussion.
In the TP53 tumor suppressor in cancer section, the authors can provide some historical background on TP53, such as when was its role as a tumor suppressor become evident from various experiments or provide some references that mentions about when it became evident that enforced expression of the wild-type TP53 protein could block oncogene-mediated transformation. Providing some background information strengthens the argument of the paper.
In the Extracellular vesicles - biogenesis, cargo and function in cancer section, the authors can add a schematic diagram of exosome and microvesicle (MV) biogenesis pathways. Maybe the authors would want to explicitly clarify that the specific sorting mechanisms might influence the composition or the subtypes of EVs released by cancer cells, and impact on the tumor-stroma communication.
In the Extracellular vesicles - function in cancer section, the authors have talked about cancer-associated fibroblasts (CAFs); I feel the paragraph is too crowded with information and can be spaced out—discuss Also it would be helpful if the authors provided some data to indicate that CAFs are a unique cell population significantly penetrating in TME and contributing to the malignant phenotype and tumorigenesis.
In the EVs as extracellular messengers of p53 section, which is the heart of the review paper, the authors have nicely articulated their argument. The authors have used a lot of previous studies to drive their point—the role of p53 in opposing epithelial-to-mesenchymal transition (EMT) in which cell–cell adhesion junctions are disassembled and cells acquire more mesenchymal, migratory phenotypes can be explained in a bit more details since EMT is tightly correlated with invasion and metastasis. Also, I feel that the section needs to be broken down into more logical sections that will make it easier for the reader to read and understand, for example, when the authors are talking about non-cell-autonomous; it would be helpful to have a sub-section, for example something like non-cell-autonomous functions of p53.
In the Conclusion section, I feel the authors are not quite conclusive and I feel it can be structured better; since I felt there wasn’t a clear and conclusive statement that wraps up the review.
Reviewer 2 Report
This is a quite nice, comprehensive review of the literature regarding p53 and extracellular vesicles (EVs). It comprises a lot of detailed information, therefore, the diagrams are very helpful to guide the reader. At the end the authors summarize brilliantly the complexity of the TP53 mutome with the challenges of pinpointing to a specific p53-evoked change in EV release.
Few minor comments:
The manuscript contains too many abbreviations which may be confusing. After a few pages, the readers outside the field might not always remember what a certain abbreviation stands for. It would be helpful to eliminate some and always write full names instead of abbreviations, like GOF, CRC, CAFs, PDAC, TME.
Page 9, line 380: "...remarkable..."
Page 10, line 408: What is "i.a."?
